# Decreasing Species Richness with Increase in Elevation and Positive Rapoport Effects of Crambidae (Lepidoptera) on Mount Taibai

**DOI:** 10.3390/insects13121125

**Published:** 2022-12-05

**Authors:** Anping Chen, Zhijie Li, Yufeng Zheng, Jinyu Zhan, Bolan Yang, Zhaofu Yang

**Affiliations:** 1Key Laboratory of Plant Protection Resources and Pest Management, Ministry of Education, Northwest A&F University, Xianyang 712100, China; 2Entomological Museum, College of Plant Protection, Northwest A&F University, Xianyang 712100, China

**Keywords:** Mount Taibai, alpha diversity, altitudinal gradient, Pyraustinae, Spilomelinae, Rapoport’s rule

## Abstract

**Simple Summary:**

Based on the investigation of altitudinal distribution data with identification by using both molecular and morphological classifications of Pyraustinae and Spilomelinae, this paper determines the altitudinal gradient pattern for these two subfamilies on the north slope of Mount Taibai of the Qinling Mountains, and provides a test of the universality of Rapoport’s rule in Lepidoptera by using four methods. Our results show that alpha diversity of Pyraustinae and Spilomelinae both decrease with rising altitude. By contrast, the species distribution ranges increase with rising altitude. Three of the four methods used to test Rapoport’s rule yielded positive results, while Rohde’s results show a unimodal distribution model and do not support Rapoport’s rule.

**Abstract:**

Rapoport’s rule proposes that a species’ range size increases with the increase in a gradient (such as latitude, altitude or water depth). However, altitudinal distributions and Rapoport’s rule have rarely been tested for Asian Lepidoptera. Pyraustinae and Spilomelinae (Lepidoptera: Crambidae) are extremely diverse in temperate Asia, including on Mount Taibai, which is considered a hotspot area for studying the vertical distribution patterns of insect species. Based on the investigation of altitudinal distribution data with identification by using both DNA barcoding and the morphological classification of Pyraustinae and Spilomelinae, this paper determines the altitudinal gradient pattern for these two subfamilies on the north slope of Mount Taibai, and provides a test of the universality of Rapoport’s rule in Lepidoptera by using four methods, including Stevens’ method, Pagel’s method, Rohde’s method, and the cross-species method. Our results show that the alpha diversity of Pyraustinae and Spilomelinae both decrease with rising altitude. By contrast, the species’ ranges increase with rising altitude. Three of the four methods used to test Rapoport’s rule yielded positive results, while Rohde’s results show a unimodal distribution model and do not support Rapoport’s rule. Our findings fill the research gap on the elevational diversity of Lepidoptera in temperate Asia.

## 1. Introduction

Species diversity is subject to the elevational gradient and constitutes the basis of studying the distribution structures of plant and animal communities [1,2]. The elevational gradient constitutes a combination of climate factors, such as light conditions, temperature, rainfall, atmospheric composition, geological constraints and other ecological factors, and the anthropization of landscapes, which affect the vertical distribution of organisms’ biodiversity [3,4,5]. Hence, the elevational pattern of species and their maintenance mechanisms provide a theoretical basis for exploring the decline of species diversity, variations in distribution, and hidden extinctions [6,7]. Since the composition of species diversity varies significantly at different elevations according to changes in environmental parameters, such as temperature and precipitation, species diversity patterns along altitudinal gradients provide crucial evidence for understanding the possible effects of climate change on biological communities [8]. Meanwhile, elevational diversity can also provide theoretical reference for researchers and relevant regulatory departments to confirm the priority of animals and plants for protection, formulate the goals of conservation priority management, and provide protection under the pressures of development [9].

Despite the altitudinal gradient pattern of species richness having been an emphasis of research over the past decades, many proposed hypotheses that determine the variation in species range size remain largely untested in most biodiversity hotspots [2,10,11]. Rapoport’s rule is an ecogeographic rule for studying the changes of biodiversity [12,13,14], asserting that the size of a species’ range is positively correlated with the relative position of the group in a gradient as the biogeographic gradient (such as latitude, altitude, or water depth) changes. It has been applied to trees, seaweeds, arthropods, marine and terrestrial molluscs, fish, amphibians, reptiles, birds, mammals, and moths [13,15]. Nonetheless, Rapoport’s rule aroused comprehensive discussion and controversy among researchers all over the world [11]. Numerous studies have confirmed Rapoport’s rule [1,16,17], though some studies have questioned its generality. For example, Beck et al. [3] clarified that temperature variability is associated with elevational species ranges as a potential gradient for supporting the mechanism of Rapoport’s rule in nocturnal moths in the Swiss Alps. By contrast, Rahbek [18] proposed that the distribution range of tropical land birds in South America on different elevation gradients did not support Rapoport’s rule. Bhattarai & Vetaas [19] studied tree species richness in the Nepalese Himalayas and found that the widest species range was at middle elevations, with narrow ranges at both the lowest and highest elevations, which is inconsistent with the Rapoport’s rule. Some scholars believe that the different verification results of Rapoport’s rule are contingent on different research objects and research scales, as well as the research hypothesis and validation methods [20,21].

Asia is the largest of Earth’s continents and contains the greatest diversity of living organisms. However, altitudinal distributions and Rapoport’s rule have rarely been tested in this region [22,23]. Mount Taibai is one of the significant dividing lines between the subtropical and warm-temperate zones in China [24]. It is also the partition line between the Palaearctic and Oriental regions in central China’s zoogeographic divisions, and has distinct climatic zones at different altitudes with rich species diversity of Lepidoptera [25,26]. As a result, it has attracted particular attention over the years as a hotspot of biodiversity in China.

In this study, we attempted to verify the universality of Rapoport’s rule from the perspective of a spatial scale by investigating the vertical distribution patterns of moths, represented by a basal, well-supported, monophyletic lineage of Crambidae, viz. Pyraustinae and Spilomelinae, on Mount Taibai. These moths have evolved to inhabit diverse ecosystems worldwide, such as arable land habitats, natural forests, piedmont plains, and high-altitude mountains [27,28,29,30,31]. The larval host associations are mainly with a variety of angiosperms, but some groups also use detritus, aquatic algae, lichens, bryophytes, ferns, and gymnosperms as their host plants [27,28,32].

Given that closely related species are very similar in morphology and appearance, species identification is extremely difficult in Lepidoptera based only on external morphological characteristics, posing problems of accuracy and speed [33,34,35]. We firstly assembled a reference DNA barcode library for the two subfamilies Pyraustinae and Spilomelinae in Crambidae to confirm species identification, and adopted four commonly used methods (Stevens’ method, Pagel’s method, Rohde’s method, and the cross-species method) to test for the impact of the Rapoport effect on species range size [13,36,37,38]. Stevens’, Pagel’s, and Rohde’s methods examine the relationship between the mean species range size and elevation, while the cross-species method estimates the relationship between altitudinal range size and the mid-point of each independent species as coordinates. We therefore hypothesize that (1) the diversity of Pyraustinae and Spilomelinae on the mountain decrease with increasing elevation, and (2) the elevation pattern of the species range size generally supports Rapoport’s rule.

## 2. Materials and Methods

### 2.1. Study Area

Mount Taibai (N 33°40′–34°10′, E 107°19′–107°58′, Figure 1) is located in Shaanxi Province on the border between Taibai County and Zhouzhi County. Its summit reaches an elevation of 3767 m and the elevation is 800 m in the foothills. This region is a transitional climate zone between subtropical and warm-temperate climates, with an average annual temperature of 1.8–2.1 °C. The average annual precipitation ranges from 800 mm to 900 mm and varies with elevation [39,40]. This region’s climate is characterized by an obvious elevation gradient with four climate zones, viz., a warm-temperate zone (800–1300 m), a temperate zone (1300–2600 m), a cool-temperate zone (2600–3350 m), and a high-elevation sub-frigid zone (3350–3767 m) [26]. The mountain contains the most abundant temperate flora, including over 1780 species of flowering plants, 325 species of bryophytes, and 110 distinct species of ferns, which can be classified into the deciduous broad-leaved forest zone (<2000 m), coniferous forest zone (2000–3000 m), alpine shrub zone (3000–3300 m), and meadow zone (>3300 m), from the foot to the top of the mountain [41,42]. It is a hotspot of biodiversity in the Oriental and Palearctic regions.

### 2.2. Sampling and Species Identification

Crambidae represents a diverse lineage occurring in most terrestrial habitats of Lepidoptera, with more than 5381 described species in the two largest subfamilies of Pyraustinae and Spilomelinae, accounting for over half of the species in the family [27,28]. Although these two subfamilies form a species-rich monophyletic basal lineage (‘PS clade’ containing Pyraustinae + Spilomelinae) in Crambidae that may represent phylogenetic and distributional congruence, the study that focuses on elevational patterns of Pyraustinae and Spilomelinae is particularly scarce in biodiversity hotspots in Asia. Through investigation of the biodiversity in the Mount Taibai Nature Reserve in the 1980s, 1435 species of insects belonging to 99 families in 19 orders were identified, including 45 species of 24 genera in Pyraustinae, and 61 species of 31 genera in Spilomelinae of the family Crambidae [25].

A total of 15 transects were established along the altitudinal gradient from 880 m to 2280 m in the northern slope of Mount Taibai at 100 m intervals. Moths of Pyraustinae and Spilomelinae were collected along the elevational gradient from a total of 30 trapping sites, including two replicate sites at each gradient, by using the automatic funnel light traps located 50 m apart (15 W ultraviolet light tube for each, wavelengths from 315 nm to 400 nm) and powered by two batteries (220 V, 85 W). The attracted moths glided down the glassy vanes through the funnel into the collecting bucket containing cyanide killing jars. Samples were collected every 30 min from 20:00–24:00, which covers the peak activity of adult crambid moths every night. The traps at each site produced three nightly catches from July to August in 2016 and 2017. No sampling sites were set above 2400 m elevations, where moths of Pyraustinae and Spilomelinaeis are unable to cope with the harsh environmental conditions. All nightly samples of two replicates at each gradient were pooled and separately placed in labeled paper bags in order to prepare pinned dry specimens. A few fresh specimens were preserved in pure ethanol.

All sampled specimens of Pyraustinae and Spilomelinae were identified and compared with material chosen by specialist Zhaofu Yang in the Entomological Museum, Northwest A&F University, Yangling, China, based on their external morphology [25,43] and particularly considering the internal structures of female and male genitalia. The following method of genitalia slide preparation was followed Yang et al. [35]. The abdomen was removed from the dried specimens and boiled in 5–10% NaOH for 5–10 min. Genitalia were stained in orange G for 15–20 min. The remainder of the insect was used for morphological and molecular analysis.

The DNA barcode molecular-based method was used in order to secure the morphology-based identification of Pyraustinae and Spilomelinae. Genomic DNA extraction was carried out using one or two legs of adult specimens and with the DNeasy DNA Extraction Kit (TransGen Biotech, Beijing, China), following the manufacturer’s protocol. The fragment of mitochondrial *COI* gene was amplified using the primers LepF1 (forward) 5′- ATTCAACCAATCATAAAGATATTGG-3′ and LepR1 (reverse) 5′-TAAACTTCTGGATGTCCAAAAAATCA-3′ [44]. PCR reactions were performed in a total volume of 25 μL using 2 μL of DNA extract, 1 µL each of forward and reverse primer, 12.5 μL of Green-Mix, and 8.5 μL of ddH_2_O. The reaction cycle consisted of pre-degeneration at 94 °C for 5 min, denaturation at 94 °C for 0.5 min, annealing at 51 °C for 1 min, extension at 72 °C for 1 min, and final extension at 72 °C for 7 min, with 40 cycles. PCR products were separated by electrophoresis in 1.0% agarose gel, and samples showing the correct band size were sequenced in both directions by AuGCT Biotech (Beijing, China). Newly sequenced 135 *COI* gene barcodes of 66 species in this study were deposited in GenBank (GenBank accessions: OK339822–OK339956). Sequences were aligned using the ClustalW default setting, and we checked their frame-shifts to avoid pseudogenes. The neighbor-joining (NJ) tree was constructed using the Kimura 2-parameter (K2P) molecular evolutionary model with 1000 bootstrap replications in MEGA X [45].

### 2.3. Alpha Diversity Analysis

To address the species richness pattern of Pyraustinae and Spilomelinae along altitudinal gradients, sampling plans were designed according to different elevations, which were divided into three altitudinal gradients based on the vertical zone of vegetation on Mount Taibai: 800–1300 m (deciduous broad-leaved forest, low altitude area), 1300–1800 m (coniferous forest, mid altitude area), and 1800–2300 m (alpine shrub, high altitude area). These divisions were made using asymptotic diversity estimates based on Hill number q [46,47]. Species richness (q = 0), Shannon diversity (q = 1), and Simpson diversity (q = 2) were calculated using the R package “iNEXT” (https://github.com/JohnsonHsieh/iNEXT, accessed on 2 November 2022) [48,49], which performed sparse extrapolation based on the relationship between the number of individuals and sampled species (data frame in Appendix A).

Species richness (q = 0) was used to estimate the presence of species. The estimation of diversity was performed using Shannon’s diversity index (q = 1), and each species’ abundance was estimated to account for the effective number of common species in the community. Simpson diversity (q = 2) was used to estimate diversity by calculating the dominant species and to predict its effective number in the community. The α diversity of two subfamilies (q = 0, 1, 2) was estimated by a 1000-bootstrap resampling with a confidence interval of 95% [47,50,51].

### 2.4. Rapoport’s Rule Analysis

Biodiversity or macroecology studies on Rapoport’s rule may lead to conflicting results due to different analytical methods and limited sampling effort in the study area [52]. Therefore, we adopted four commonly used methods to test for the impact of the Rapoport effect on species range size of representative species (remaining species represented by a single specimen were ruled out due to their undetermined elevational range size) for both Pyraustinae and Spilomelinae, including Stevens’ method [13], Pagel’s method [36], Rohde’s method [37], and the cross-species method [38].

For Stevens’ method, the altitudinal Mount Taibai gradient was divided into 200 m vertical elevation bands. The mean altitudinal range of each species was calculated by averaging the elevational ranges present in the band [16]. Pagel’s method quantified the average species range size of all species whose upper distribution limit fell within a given latitudinal band [36]. Rohde’s method developed by Rohde et al. [37] uses each species as an independent unit from which the mid-point between the lowest and highest elevational records is calculated. In the cross-species method, a scatter plot was made with altitudinal range size and the mid-point of each independent species as coordinates. The model slopes of the four methods were obtained using fitting linearity to test the existence of Rapoport’s effect [53].

### 2.5. Statistical Analysis

The Pearson correlation coefficients between bivariate variables were determined using SPSS 22.0 (SPSS, Chicago, IL, USA), including mean species range and elevation (Appendix A).

## 3. Results

### 3.1. Species Delimitation

A total of 1143 individuals representing 73 species of 42 genera in Pyraustinae and Spilomelinae of Crambidae were collected on Mount Taibai (Appendix A). We identified 70 species, which represented 95.89% of all sampled specimens. The remaining three morphospecies were identified at genus level. For DNA barcode identification, we sequenced 135 *COI* gene barcodes of 66 species in 34 genera (GenBank accessions: OK339822–OK339956, see Appendix A). Our results showed that all barcoded species can be accurately assigned into two corresponding subfamilies in the Neighbour-joining (NJ) tree (Appendix A), representing Pyraustinae and Spilomelinae. The subfamily Pyraustinae was divided into six clades: *Sitochroa*, *Circobotys*, *Anania*, *Ostrinia*, *Protcurhypara* and *Opsibotys*. In addition, the Spilomelinae formed 57 clades representing 57 species in 28 genera. In general, all 66 barcoded species in 34 genera branched independently and each of them formed monophyletic clades, indicating that the success of molecular identification based on *COI* gene barcodes was 100%.

### 3.2. Alpha Diversity along Altitude

In this study, a sample completeness curve (Figure 2) was generated to estimate whether the sampling effort is sufficient along the altitudinal gradient for Pyraustinae and Spilomelinae on Mount Taibai. The results showed that the curves rose sharply and then gradually flattened out, with the sample coverage rate reaching over 90%, indicating that sufficient sampling was achieved along the low-altitude area (<1300 m), mid-altitude area (1300–1800 m), and high altitude area (>1800 m). However, the sample completeness for the low/mid-altitude areas was estimated to be higher than that of the high-altitude area.

The alpha diversity analysis (Figure 3) showed that the species diversity of Pyraustinae and Spilomelinae decreased with increasing altitude in order of “q” (0–2), although the cumulative species curves did not approach an asymptote. The species richness values were 57, 52, and 38 for the low-, mid-, and high-altitude areas, respectively; the Shannon diversity values were 33.1, 27.2, and 19.9; and the Simpson diversity values were 24.3, 17.3, and 13.6. The low-altitude area has the highest species diversity and effective number of common species and dominant species, followed by the middle altitude area, and the high-altitude area has the lowest species diversity.

The vast majority of common species belong to the subfamily Spilomelinae, except for two members of Pyraustinae, viz. Anania verbascalis and Pyrausta noctualis. The most dominant species in the study area was *Goniorhynchus marginalis*, contributing 11.29% of the total number of sampled crambid moths on Mount Taibai. *Syllepte invalidalis* and *Tyspanodes hypsalis* dominated samples below 1300 m (mid-point of altitude), where these two species accounted for 5.77% and 3.85% of total individuals, respectively. *G. marginalis* dominated the area between 1300 m to 1800 m, where it accounted for 11.29%. Above 1800 m, *Udea costalis* dominated the samples (1.84%).

### 3.3. Elevational Distribution Ranges and Rapoport’s Rule Test

The mean elevational species range in our samples was 654.12 m for 59 representative species out of 73 (remaining 14 species represented by a single specimen were excluded). Many species had a much more limited distribution range, with 43 out of 59 species (72.88%) recorded at less than 1000 m. The maximal altitudinal range (1333 m) was observed in *Goniorhynchus marginalis*, *Palpita nigropunctalis*, and *Pycnarmon cribrata*, while the minimal altitudinal range (67–200 m) was observed in *Spoladea recurvalis*, *Ostrinia scapulalis*, *Proteurrhypara cuspidata*, *Sitochroa palealis*, *Agrotera nemoralis*, *Conogethes punctiferalis*, *Mecyna gracilis*, *Nosophora* sp., *Udea* sp., *Omiodes tristrialis*, *Syllepte taiwannalis*, *Tabidia strigiferalis*, and *Udea costalis*.

To test for the Rapoport effect on species range size, four methods were used in this study, based on the 59 representative species of Pyraustinae and Spilomelinae on Mount Taibai. Three of the four methods used to test Rapoport’s rule yielded positive results. Stevens’ method indicated a positive linear relationship between the mean altitudinal range and the elevation on Mount Taibai (R^2^ = 0.8207, Figure 4a; *p* < 0.005). With the gradual increase in altitude, the average altitude distribution width of two subfamilies in every 200 m altitude section generally showed a steadily increasing trend from 900 m to 2300 m. That means that altitudinal ranges increased as elevations increased, despite a minor decrease at the 1700 m to 1900 m section. The results obtained by Stevens’ method support Rapport’s rule.

The results of Pagel’s method’s showed that the average species width pattern of the species was an upward trend along with increasing altitude. At the 900 m to 1100 m altitude section, the mean species range was the lowest, which is similar to the results from Stevens’ method. For another five elevations (1300 m to 2100 m), the species distribution width of the two subfamilies increased linearly with increasing altitude, and reached the highest distribution width at the 2100 m to 2300 m section. Therefore, the test results of Pagel’s method (R^2^ = 0.9968, Figure 4b; *p* < 0.001) also support Rapoport’s rule.

Rohde’s method showed that the species distribution widths of Pyraustinae and Spilomelinae first increased and then decreased with the increase in altitude. A unimodal pattern was evident in a parabolic regression made using the mid-point method, and the maximum value was observed at the 1500 m to 1700 m section. The results of the mid-point method (R^2^ = 0.7411, Figure 4c; *p* < 0.727) do not support Rapoport’s rule, which is likely related to the limitations of the mid-domain effect.

The cross-species method indicated that the relationship between the width of the species domain and the altitude gradient of the subfamily Pyraustinae and Spilomelinae is roughly triangular. The result of linear fitting (R^2^ = 0.0563, Figure 4d; *p* < 0.07) supported Rapoport’s rule.

## 4. Discussion

### 4.1. Efficiency of Species Recognition

In practice, accurate species identification plays a very important role in species diversity monitoring [54]. In this study, 66 representative species of 34 genera in two subfamilies of lepidopteran insects were studied by using DNA barcoding and morphological taxonomy to explore the altitudinal diversity and to test Rapoport’s rule on Mount Taibai. All sequences generated were longer than 650 bp in length, and there were no insertions or deletions, which met the data standard of full-length DNA barcoding sequences and could be applied to the molecular identification of the two subfamilies [34,55]. Furthermore, we also combined our analysis with information about the species’ genital features and external morphological characteristics to ensure the identification results of species are reliable and accurate, which is corroborated by findings in many previous studies [34,56]. However, the species identification is unreliable in many studies because they do not integrate both morphological and molecular approaches [55]. Our findings confirm that DNA barcoding is a complementary method to morphology-based identification, and assists in delimiting morphologically indistinguishable species [55,56,57]. In species-rich hotspots that are largely unexplored, DNA barcoding will play an important role for assessing and understanding global biodiversity [58].

### 4.2. Alpha Diversity with Increasing Altitude

Our results highlight that α diversity of the subfamilies Pyraustinae and Spilomelinae on Mount Taibai (Figure 3) gradually decreases with increasing altitude and is somehow conformed to a monotonic decreasing pattern, which are consistent with trends observed in the vegetation communities [59] and for geometrids [60] that both show a general species richness decrease with increasing elevation on Changbai Mountain, Northeast China. A strong decrease in invertebrate species’ abundance with increasing altitude has similarly been reported from tropical forests [61]. Similarity, Fiedler et al. [62] found that there was a steep and steady decline in species diversity of some crambids from about 1000–2700 m a.s.l from a humid tropical mountain in South America. However, Beck et al. [4] illuminated that global species richness in geometrid moths was characterized by midpeak patterns along major elevational gradients.

The decreasing species diversity pattern of Pyraustinae and Spilomelinae is likely a result of the unique climate environment and vegetation habitats on Mount Taibai, in particular along altitudes of 800–3500 m [26]. Compared with coniferous forests and alpine shrubs at higher elevations, the deciduous broad-leaved forest contains diverse vegetation and complex habitats for the survival of these crambids [63]. For example, host plant generalists, such as *Goniorhynchus* spp., *Palpita* spp., and *Pycnarmon cribrata*, enable these two subfamilies to inhabit many more habitats and elevational distribution ranges on Mount Taibai than some specialists (i.e., *Ostrinia scapulalis*, *Sitochroa palealis*, *Conogethes punctiferalis*, *Mecyna gracilis*, *Udea* spp., etc.) possessing a narrow host plant spectrum, which is consistent with the distributional pattern observed in European mountainous areas [64,65,66].

In addition, temperature and precipitation conditions might be responsible for the decreasing pattern of species richness in the two subfamilies, suggesting that environmental predictors (temperature and precipitation) are key factors for the survival, reproduction, and distribution boundaries of insects [67]. Similarly, previous studies indicated that disturbances caused by human activities, the fragmentation of landscape and habitats, and the reduction of natural predators in low-altitude areas could significantly influence the insect abundance [6,68,69], which is corroborated by our results. In other words, the sampled species and individuals at an 800 m altitude (excluded from the entire elevational gradients) were much less numerous than those in other altitude sampling sites. However, the limited sample size along the altitudinal gradient may have impacted the prediction of the diversity of the two subfamilies and resulted in the advent of outliers in the present study. Improving sampling efforts to ensure there are sufficient materials at each elevational gradient, particularly at high altitudes, would undoubtedly reduce the outliers in future studies.

### 4.3. Rapoport’s Rule Test and Factors Influencing Elevational Species Range Patterns

Rapoport’s rule proposes that species richness gradually decreases with the increase in altitude, and the width of a species range increases instead [12]. Our results show that Stevens’ method, Pagel’s method, and cross-species method generally support Rapoport’s rule, while the results of the mid-point method do not support Rapoport’s rule. A linear trend cannot be obtained by using the mid-point method, which is consistent with previous research results [21]. This may be affected by a mid-domain effect in which only the narrow-range species are included at both ends of the distribution gradient, while the species with the widest ranges are included only in the middle of the distribution [37,70,71]. Thus, it is easy to show a single peak curve pattern with the increase of the species range width, and a decline with the increase of the altitude (the peak position is due to the location of the point varying in the distribution pattern of the environmental gradient) [70]. Despite the results of Rohde’s method violating Rapoport’s rule, all three other methods illuminate that species’ altitudinal ranges within the two subfamilies of Crambidae on Mount Taibai positively correlate with elevation with higher coefficients, providing a theoretical basis for revealing the altitudinal distribution pattern of Lepidoptera in this region, and providing a scientific basis for large-scale biodiversity conservation. Besides, the small sample size along the altitudinal gradient might have impacted the predictions of Rapoport’s rule using different methods. Further investigations based on much more comprehensive sampling representatives of Lepidoptera are essential to reveal their elevational patterns on Mount Taibai.

Moreover, our findings suggest that the Ambient Energy Hypothesis may result in the significant linear increase in species’ altitudinal ranges with increasing elevation observed in this study [72]. The results presented here are also likely related to the hypothesis that the climate tolerance of higher-altitudes species is more extensive than that of lower-altitudes species [13].

## 5. Conclusions

In conclusion, the results in the present study confirm that integrated approaches to assemble more evidence for accelerating species delimitation will facilitate the surveillance of native biodiversity hotspots in temperate Asia. Meanwhile, we conclude that the altitudinal pattern of diversity we describe for Pyraustinae and Spilomelinae supports the predictions of Rapoport’s effect that diversity should be greatest at lower elevations, while species altitudinal ranges should be broader at higher elevations.

## Figures and Tables

**Figure 1 insects-13-01125-f001:**
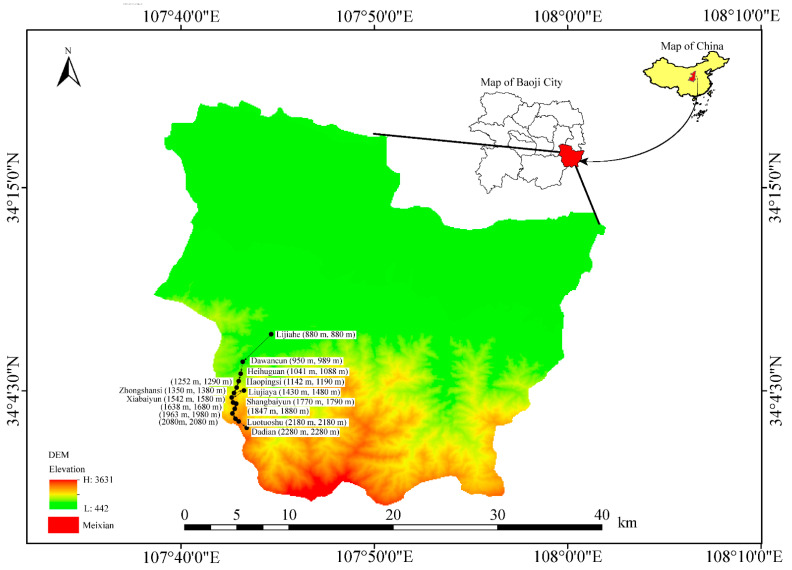
Sampling localities on Mount Taibai. DEM (Digital Elevation Model) downloaded from the Geospatial Data Cloud (http://www.gscloud.cn/search, accessed on 2 November 2022). Solid circle in black indicates sampling sites at different elevations. Elevations in parentheses represent two replicates at same transect. Named sampling sites were labelled directly in the map.

**Figure 2 insects-13-01125-f002:**
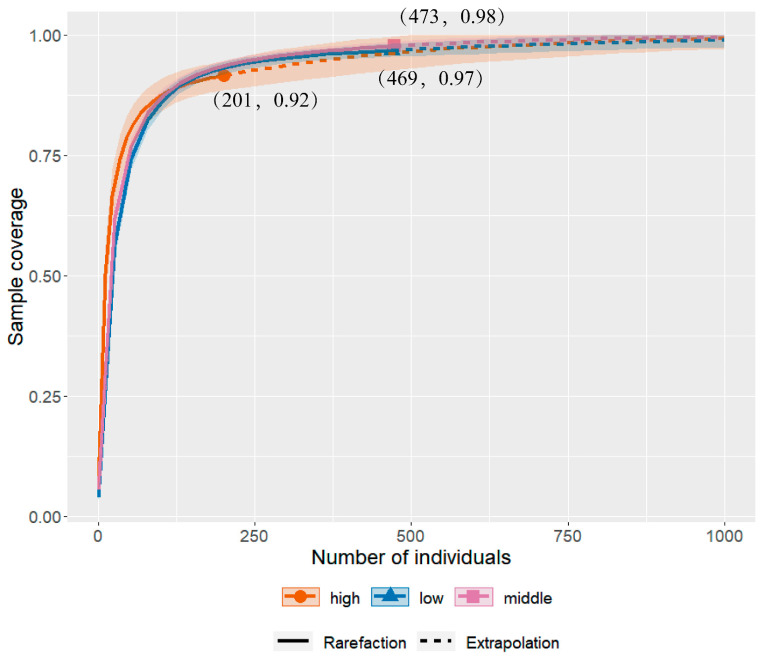
Sample completeness curve of Pyraustinae and Spilomelinae among different altitudes.

**Figure 3 insects-13-01125-f003:**
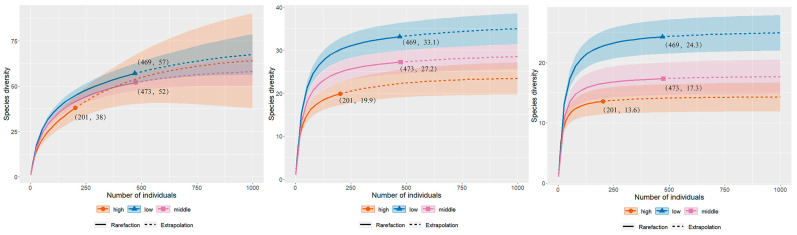
Sample-size-based rarefaction (solid line) and extrapolation (dotted line) sampling curves with 95% confidence intervals (shaded areas) for three different elevations, separately by diversity order: q = 0 (species richness, left panel), q = 1 (Shannon diversity, middle panel) and q = 2 (Simpson diversity, right panel). The solid dots/triangles/squares represent the reference samples.

**Figure 4 insects-13-01125-f004:**
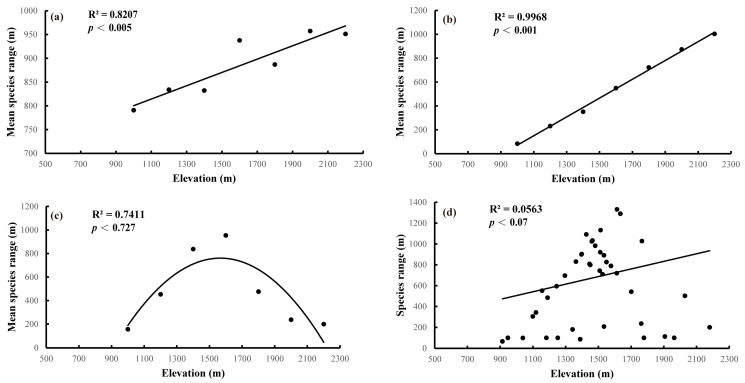
Results of four methods to test Rapoport’s effect. (**a**) Stevens’ method, (**b**) Pagel’s method, (**c**) Mid-point method, and (**d**) cross-species method.

## Data Availability

Appendices in the study are available in Dryad (https://doi.org/10.5061/dryad.xd2547djm). Sequences data generated in this study are openly available in GenBank at https://www.ncbi.nlm.nih.gov, accessed on 30 October 2022 (GenBank accessions: OK339822–OK339956).

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
