# Peer review of "Decreasing Species Richness with Increase in Elevation and Positive Rapoport Effects of Crambidae (Lepidoptera) on Mount Taibai"

_insects, 2022, doi:10.3390/insects13121125_

Round 1

Reviewer 1 Report

This paper investigates the vertical distribution patterns of two subfamilies of Crambidae : Pyraustinae and Spilomelinae, along an altitudinal transect on Mount Taibai. Authors used a combination of both morphology and DNA barcoding to identify the collected moths. They analysed the data using several methods to test whether the elevation pattern of species range size follows the Rapoport’s rule. The paper represents an important contribution to the altitudinal gradient ecology in general and moth ecology in particular.  The data seems to support their conclusions. Please see below some comments for authors to consider:

1.     Authors have deposited the newly generated COI sequences in Genbank which is great however the sequences will not be associated to their metadata (photo of each sequenced voucher and their collecting data). I would therefore strongly recommend authors to deposit their barcodes in the online barcoding data base BOLD (http://www.boldsystems.org/) with all metadata for each specimens analysed (georeferenced collecting data,…) and create a data set in BOLD and include its DOI number in the main text. To see how to create a data set and obtain a DOI please check page 26 of the following online manual :

2.     Line 224 : once your DNA barcodes have been  deposited in BOLD please state how many of those can be identified with BOLD down to species level and how many are new to BOLD. Also, how many Barcode Index Numbers (BINs) are associated to the data set. Please also state how many species have multiple BINs if any.

3.     Authors should include a table summarizing the specimens used in the study. Indicatingwhich ones have been identified only morphologically and those which have been identified using DNA barcode data. For the former please include data such as the number of the genital preparation and for the latter, include process IDs (=barcode numbers), genbank accession numbers, BINs. You can follow the supplementary table in the following paper as a model.

2016 Huemer, P., Lopez-Vaamonde, C and Triberti, P. A new genus and species of leaf-mining moth from the French Alps, Mercantouria neli gen. n., sp. n. (Lepidoptera: Gracillariidae). Zookeys 586 : 145-162

Reviewer 2 Report

The authors investigated community structure and species range of Crambidae family in mount Tabai China. This is an interesting and well designed study of not well-known taxa. 

Given that this study has great entomological purpose, I suggest to the authors to more introduce ecological features of Crambidae in introduction and especially in discussion for some "particular" species that has been identify in this study. These adds could be interesting for a broader range of reader in the scope of Insects' journal.

In abstract and simple summary : please briefly explain the Rapoport effect.

Ln 27: change "traditionnal classification" to "morphological identification" which is a more precise term

Ln 42-45: It could be interesting to introduce the effect of the Anthropization of landscape even in mountains on the biodiversity. I would think that this is more impactfull on the community structure for the Lepidopteran biodiversity in mountains.

Ln 52: "elevational"

Ln 87: What you mean by "('PS clade')" ?

Ln 102-107 : Does the "(Stevens’s method, Pagel’s method,Rohde’s method, and the cross-species method)" has methodological references or a review of these ? If yes, you could add this here.

Ln 147-170: Did you compare your specimens to a reference collection ? Did you have at least one taxonomist specialized in these Crambidae sub-family ? If yes, please mention it, it is important to certify that you identify well the species.

Ln 156-157: Why this information is important in the manuscript "Specimens were selected from different altitudes." ?

Ln 174-179: Why you did not classify the gradient of sampling such as it was showed in M&M as it is exposed to Figure 2 ? How do you take account of the bias introduce by the unbalanced (in term of sampling effort) altitudinal gradients: 900 to 1300m (400m);1300 to 1900 (600m);1900m to 2400m (400m) ? 

Ln 177-179: Explain better why you choose the Hill framework to estimate your alpha diversity index ?

Ln 180-184: 

This paper presents an interesting study, but the use of Alpha Diversity Indices presents serious conceptual and statistical problems,  which make comparisons of species richness or species abundances across

communities nearly impossible. I present below the best solution for solve  the problem.

The problems: Both Shannon and Simpson Indices are the sum of calculations (number of species) and consequently are biased by sampling effort.

The solution: The best way forward is the application of Hill Numbers  (see Jost, 2006; and Tuomisto, 2010), that provide a framework to integrate the previous concepts into a consistent terminology,

        based on  the paper by Hill (1973). The Hill numbers, also called the “effective number of species”, as the best choice to quantify abundance-based species diversity. Here, the effective number of species means the

         number of equally abundant species that are needed to give the same   value of a diversity measure.

         Therefore, the best solution is to use the sequence o Hill Numbers as follows: N0 = species richness (S); N1= Exponential

        Shannon-Wiener (exp H´); N2 Inverse Simpson´s index (1/D); N4 = Berger-Parker index (1/d)(see also Magurran 2004).

         The reason for applying the Hill numbers are due to:

         Problem 1: The replication principle, in which imagine two

        communities with four species and four individuals C1 = { 1 1 1 1 } and with

        eight species and eight individuals C2 = { 1 1 1 1 1 1 1 1 }

         The results will be:

         For C1

         S = 4

         D = 0.75

         1/D = 4

         For C2

         S = 8

         D = 0.875

         1/D = 8

         This implies that there is mo replication for Gini-Simpson (D),

        but it is OK for Inverse-Simpson (1/D)

         Problem 2: The linearity principle Absence of linearity for Shannon (H) & Simpson (D), but OK for Inverse-Simpson (1/D)

         Suppose you are comparing the diversity of aquatic microorganisms

        before and after an oil spill (L. Jost)

         Before: Gini-Simpson = 0.99

         After: Gini-Simpson = 0.97

         you conclude that the magnitude of the drop is small. You might

        even say

         (very wrongly) that the diversity has dropped by 2%, which sounds

        like a

         small drop, nothing to worry about.

         IF YOU APPLY CORRECTLY the Hill Numbers:

         Before: Inverse-Simpson = 100 (1/1-0.99)

         After: Inverse-Simpson= 33 (1/1-0.97)

         The difference between the pre- and post-spill diversities is in fact

         enormous. The drop in diversity is 66%, not 2%!

         References

         Hill M. O. 1973. Diversity and evenness: a unifying notation and its

         consequences. Ecology 54, 427–43210.2307/1934352

        (doi:10.2307/1934352)

         Jost L. 2006. Entropy and diversity. Oikos 113,

         363–37510.1111/j.2006.0030-1299.14714.x

         (doi:10.1111/j.2006.0030-1299.14714.x)

         Magurran, A. E. 2004. Measuring Ecological Diversity. Blackwell

         Publishing, Oxford. 256 pp.

         Tuomisto H. 2010. A consistent terminology for quantifying species

         diversity? Yes, it does exist. Oecologia 164,

         853–86010.1007/s00442-010-1812-0 (doi:10.1007/s00442-010-1812-0)

         ----------------------------------------------------

Ln 186-189: For what scientific reason do you classify species for common and uncommon ones ? (Best is to support it with a reference)

Ln 206-210 + Appendix S1-S2-S3: How do you obtain you R² ? To what coefficient this corresponds ?

Ln237-249 : Test it with Hill-Number index

Figure 5: Why there is no regression line in figure 5c ?

Reviewer 3 Report

The manuscript is very interested, all the aspects have been thoroughly explored in the manuscript and well defined with appropriate hyphotesys and reference to support. Even the statistical analyses have been done appropriately and meticulously. It is a well-balanced manuscript, and with an good discussion.

I found only some mistakes in the references part: line 471, 483, 489 503, 535. Please correct them. 

I believe that the manuscript is to be accepted without revision. 

Round 2

Reviewer 2 Report

Thanks for the improvement of the manuscript. For me it is OK in the present form except for the image resolution of Figure 3 which to improve.

Kinds regards 
